# Spectral Learning of Mixture of Hidden Markov Models

**Y. Cem Sübakan**[b], **Johannes Traa**[♯], **Paris Smaragdis**[b,♯,♮]
[b]Department of Computer Science, University of Illinois at Urbana-Champaign
[♯]Department of Electrical and Computer Engineering, University of Illinois at Urbana-Champaign
[♮]Adobe Systems, Inc.
{subakan2, traa2, paris}@illinois.edu

## Abstract

In this paper, we propose a learning approach for the Mixture of Hidden Markov Models (MHMM) based on the Method of Moments (MoM). Computational advantages of MoM make MHMM learning amenable for large data sets. It is not possible to directly learn an MHMM with existing learning approaches, mainly due to a permutation ambiguity in the estimation process. We show that it is possible to resolve this ambiguity using the spectral properties of a global transition matrix even in the presence of estimation noise. We demonstrate the validity of our approach on synthetic and real data.

## 1 Introduction

Method of Moments (MoM) based algorithms [1, 2, 3] for learning latent variable models have recently become popular in the machine learning community. They provide uniqueness guarantees in parameter estimation and are a computationally lighter alternative compared to more traditional maximum likelihood approaches. The main reason behind the computational advantage is that once the moment expressions are acquired, the rest of the learning work amounts to factorizing a moment matrix whose size is independent of the number of data items. However, it is unclear how to use these algorithms for more complicated models such as Mixture of Hidden Markov Models (MHMM).

MHMM [4] is a useful model for clustering sequences, and has various applications [5, 6, 7]. The E-step of the Expectation Maximization (EM) algorithm for an MHMM requires running forward-backward message passing along the latent state chain for each sequence in the dataset in every EM iteration. For this reason, if the number of sequences in the dataset is large, EM can be computationally prohibitive.

In this paper, we propose a learning algorithm based on the method of moments for MHMM. We use the fact that an MHMM can be expressed as an HMM with block diagonal transition matrix. Having made that observation, we use an existing MoM algorithm to learn the parameters up to a permutation ambiguity. However, this doesn't recover the parameters of the individual HMMs. We exploit the spectral properties of the *global* transition matrix to estimate a de-permutation mapping that enables us to recover the parameters of the individual HMMs. We also specify a method that can recover the number of HMMs under several spectral conditions.

## 2 Model Definitions

### 2.1 Hidden Markov Model

In a Hidden Markov Model (HMM), an observed sequence $\mathbf{x} = x_{1:T} = \{x_1, \ldots, x_t, \ldots, x_T\}$ with $x_t \in \mathbb{R}^L$ is generated conditioned on a latent Markov chain $\mathbf{r} = r_{1:T} = \{r_1, \ldots, r_t, \ldots, r_T\}$, with

$r_t \in \{1, \ldots M\}$. The HMM is parameterized by an emission matrix $O \in \mathbb{R}^{L \times M}$, a transition matrix $A \in \mathbb{R}^{M \times M}$ and an initial state distribution $\nu \in \mathbb{R}^M$. Given the model parameters $\theta = (O, A, \nu)$, the likelihood of an observation sequence $x_{1:T}$ is defined as follows:

$$p(x_{1:T}|\theta) = \sum_{r_{1:T}} p(x_{1:T}, r_{1:T}|\theta) = \sum_{r_{1:T}} \prod_{t=1}^{T} p(x_t|r_t, O)\, p(r_t|r_{t-1}, A)$$

$$= 1_M^\top A \operatorname{diag}(p(x_T| :, O)) \cdots A \operatorname{diag}(p(x_1| :, O))\, \nu = 1_M^\top \left( \prod_{t=1}^{T} A\operatorname{diag}(O(x_t)) \right) \nu, \quad (1)$$

where $1_M \in \mathbb{R}^M$ is a column vector of ones, we have switched from index notation to matrix notation in the second line such that summations are embedded in matrix multiplications, and we use the MATLAB colon notation to pick a row/column of a matrix. Note that $O(x_t) := p(x_t| :, O)$. The model parameters are defined as follows:

- $\nu(u) = p(r_1 = u|r_0) = p(r_1 = u)$      initial latent state distribution
- $A(u, v) = p(r_t = u|r_{t-1} = v),\ t \geq 2$      latent state transition matrix
- $O(:, u) = \mathbb{E}[x_t|r_t = u]$      emission matrix

The choice of the observation model $p(x_t|r_t)$ determines what the columns of $O$ correspond to:

- Gaussian: $p(x_t|r_t = u) = \mathcal{N}(x_t; \mu_u, \sigma^2)$    $\Rightarrow$    $O(:, u) = \mathbb{E}[x_t|r_t = u] = \mu_u.$
- Poisson: $p(x_t|r_t = u) = \mathcal{PO}(x_t; \lambda_u)$    $\Rightarrow$    $O(:, u) = \mathbb{E}[x_t|r_t = u] = \lambda_u.$
- Multinomial: $p(x_t|r_t = u) = \textit{Mult}(x_t; p_u, S)$    $\Rightarrow$    $O(:, u) = \mathbb{E}[x_t|r_t = u] = p_u.$

The first model is a multivariate, isotropic Gaussian with mean $\mu_u \in \mathbb{R}^L$ and covariance $\sigma^2 I \in \mathbb{R}^{L \times L}$. The second distribution is Poisson with intensity parameter $\lambda_u \in \mathbb{R}^L$. This choice is particularly useful for counts data. The last density is a multinomial distribution with parameter $p_u \in \mathbb{R}^L$ and number of draws $S$.

## 2.2 Mixture of HMMs

The Mixture of HMMs (MHMM) is a useful model for clustering sequences where each sequence is modeled by one of $K$ HMMs. It is parameterized by $K$ emission matrices $O_k \in \mathbb{R}^{L \times M}$, $K$ transition matrices[1] $A_k \in \mathbb{R}^{M \times M}$, and $K$ initial state distributions $\nu_k \in \mathbb{R}^M$ as well as a cluster prior probability distribution $\pi \in \mathbb{R}^K$. Given the model parameters $\theta_{1:K} = (O_{1:K}, A_{1:K}, \nu_{1:K}, \pi)$, the likelihood of an observation sequence $\mathbf{x}_n = \{x_{1,n}, x_{2,n}, \ldots, x_{T_n,n}\}$ is computed as a convex combination of the likelihood of $K$ HMMs:

$$p(\mathbf{x}_n|\theta_{1:K}) = \sum_{k=1}^{K} p(h_n = k)p(\mathbf{x}_n|h_n = k, \theta_k) = \sum_{k=1}^{K} \pi_k \sum_{r_{1:T_n,n}} p(\mathbf{x}_n, \mathbf{r}_n|h_n = k, \theta_k)$$

$$= \sum_{k=1}^{K} \pi_k \sum_{r_{1:T_n,n}} \prod_{t=1}^{T_n} p(x_{t,n}|r_{t,n}, h_n = k, O_k)p(r_{t,n}|r_{t-1,n}, h_n = k, A_k)$$

$$= \sum_{k=1}^{K} \pi_k \left\{ 1_J^\top \left( \prod_{t=1}^{T_n} A_k \operatorname{diag}(O_k(x_{t,n})) \right) \nu_k \right\}, \quad (2)$$

where $h_n \in \{1, 2, \ldots, K\}$ is the latent cluster indicator, $\mathbf{r}_n = \{r_{1,n}, r_{2,n}, \ldots, r_{T_n,n}\}$ is the latent state sequence for the observed sequence $\mathbf{x}_n$, and $O_k(x_{t,n})$ is a shorthand for $p(x_{t,n}| :, h_n = k, O_k)$. Note that if a sequence is assigned to the $k^{\text{th}}$ cluster ($h_n = k$), the corresponding HMM parameters $\theta_k = (A_k, O_k, \nu_k)$ are used to generate it.

# 3 Spectral Learning for MHMMs

Traditionally, the parameters of an MHMM are learned with the Expectation-Maximization (EM) algorithm. One drawback of EM is that it requires a good initialization. Another issue is its computational requirements. In every iteration, one has to perform forward-backward message passing for every sequence, resulting in a computationally expensive process, especially when dealing with large datasets.

The proposed MoM approach avoids the issues associated with EM by leveraging the information in various moments computed from the data. Given these moments, which can be computed efficiently, the computation time of the learning algorithm is independent of the amount of data (number of sequences *and* their lengths).

Our approach is mainly based on the observation that an MHMM can be seen as a single HMM with a block-diagonal transition matrix. We will first establish this proposition and discuss its implications. Then, we will describe the proposed learning algorithm.

## 3.1 MHMM as an HMM with a special structure

**Lemma 1:**

An MHMM with *local* parameters $\theta_{1:K} = (O_{1:K}, A_{1:K}, \nu_{1:K}, \pi)$ is an HMM with *global* parameters $\bar{\theta} = (\bar{O}, \bar{A}, \bar{\nu})$, where:

$$\bar{O} = [O_1 \ O_2 \ \ldots \ O_K] \quad , \quad \bar{A} = \begin{bmatrix} A_1 & 0 & \ldots & 0 \\ 0 & A_2 & \ldots & 0 \\ & & \ddots & \\ 0 & 0 & \ldots & A_K \end{bmatrix} \quad , \quad \bar{\nu} = \begin{bmatrix} \pi_1 \nu_1 \\ \pi_2 \nu_2 \\ \vdots \\ \pi_K \nu_K \end{bmatrix} . \tag{3}$$

**Proof:** Consider the MHMM likelihood for a sequence $\mathbf{x}_n$:

$$p(\mathbf{x}_n | \theta_{1:K}) = \sum_{k=1}^{K} \pi_k \left\{ 1_M^\top \left( \prod_{t=1}^{T_n} A_k \operatorname{diag}(O_k(x_t)) \right) \nu_k \right\} \tag{4}$$

$$= 1_{MK}^\top \left( \prod_{t=1}^{T_n} \begin{bmatrix} A_1 & 0 & \ldots & 0 \\ 0 & A_2 & \ldots & 0 \\ & & \ddots & \\ 0 & 0 & \ldots & A_K \end{bmatrix} \operatorname{diag}([O_1 \ O_2 \ \ldots \ O_K](x_t)) \right) \begin{bmatrix} \pi_1 \nu_1 \\ \pi_2 \nu_2 \\ \vdots \\ \pi_K \nu_K \end{bmatrix}$$

$$= 1_{MK}^\top \left( \prod_{t=1}^{T_n} \bar{A} \operatorname{diag}(\bar{O}(x_t)) \right) \bar{\nu},$$

where $[O_1 \ O_2 \ \ldots \ O_K](x_t) := \bar{O}(x_t)$. We conclude that the MHMM and an HMM with parameters $\bar{\theta}$ describe equivalent probabilistic models. $\square$

We see that the state space of an MHMM consists of $K$ disconnected regimes. For each sequence sampled from the MHMM, the first latent state $r_1$ determines what region the entire latent state sequence lies in.

## 3.2 Learning an MHMM by learning an HMM

In the previous section, we showed the equivalence between the MHMM and an HMM with a block-diagonal transition matrix. Therefore, it should be possible to use an HMM learning algorithm such as spectral learning for HMMs [1, 2] to find the parameters of an MHMM. However, the true global parameters $\bar{\theta}$ are recovered inexactly due to noise $\epsilon$: $\bar{\theta} \to \bar{\theta}_\epsilon$ and state indexing ambiguity via a permutation mapping $\mathcal{P}$: $\bar{\theta}_\epsilon \to \bar{\theta}_\epsilon^{\mathcal{P}}$. Consequently, the parameters $\bar{\theta}_\epsilon^{\mathcal{P}} = (\bar{O}_\epsilon^{\mathcal{P}}, \bar{A}_\epsilon^{\mathcal{P}}, \bar{\nu}_\epsilon^{\mathcal{P}})$ obtained from the learning algorithm are in the following form:

$$\bar{O}_\epsilon^{\mathcal{P}} = \bar{O}_\epsilon P^\top, \quad \bar{A}_\epsilon^{\mathcal{P}} = P \bar{A}_\epsilon P^\top, \quad \bar{\nu}_\epsilon^{\mathcal{P}} = P \bar{\nu}_\epsilon , \tag{5}$$

where $P$ is the permutation matrix corresponding to the permutation mapping $\mathcal{P}$.

The presence of the permutation is a fundamental nuisance for MHMM learning since it causes parameter mixing between the individual HMMs. The global parameters are permuted such that it becomes impossible to identify individual cluster parameters. A brute force search to find $\mathcal{P}$ requires $(MK)!$ trials, which is infeasible for anything but very small $MK$. Nevertheless, it is possible to efficiently find a depermutation mapping $\widetilde{\mathcal{P}}$ using the spectral properties of the global transition matrix $\bar{A}$. Our ultimate goal in this section is to undo the effect of $\mathcal{P}$ by estimating a $\widetilde{\mathcal{P}}$ that makes $\bar{A}_\epsilon^{\mathcal{P}}$ block diagonal despite the presence of the estimation noise $\epsilon$.

### 3.2.1 Spectral properties of the global transition matrix

**Lemma 2:**

Assuming that each of the local transition matrices $A_{1:K}$ has only one eigenvalue which is 1, the global transition matrix $\bar{A}$ has $K$ eigenvalues which are 1.

**Proof:**

$$
\bar{A} = \begin{bmatrix} V_1\Lambda_1 V_1^{-1} & \cdots & 0 \\ 0 & \ddots & 0 \\ 0 & 0 & V_K\Lambda_K V_K^{-1} \end{bmatrix} = \underbrace{\begin{bmatrix} V_1 & \cdots & 0 \\ 0 & \ddots & 0 \\ 0 & 0 & V_K \end{bmatrix} \begin{bmatrix} \Lambda_1 & \cdots & 0 \\ 0 & \ddots & 0 \\ 0 & 0 & \Lambda_K \end{bmatrix} \begin{bmatrix} V_1 & \cdots & 0 \\ 0 & \ddots & 0 \\ 0 & 0 & V_K \end{bmatrix}^{-1}}_{\bar{V}\bar{\Lambda}\bar{V}^{-1}},
$$

where $V_k\Lambda_k V_k^{-1}$ is the eigenvalue decomposition of $A_k$ with $V_k$ as eigenvectors, and $\Lambda_k$ as a diagonal matrix with eigenvalues on the diagonal. The eigenvalues of $A_{1:K}$ appear unaltered in the eigenvalue decomposition of $\bar{A}$, and consequently $\bar{A}$ has $K$ eigenvalues which are 1. $\qquad\square$

**Corollary 1:**

$$
\lim_{e\to\infty} \bar{A}^e = \begin{bmatrix} \bar{v}_1 1_M^\top & \cdots & \bar{v}_k 1_M^\top & \cdots & \bar{v}_K 1_M^\top \end{bmatrix}, \tag{6}
$$

where $\bar{v}_k = [0^\top \ \cdots \ v_k^\top \ \cdots \ 0^\top]^\top$ and $v_k$ is the stationary distribution of $A_k, \forall k \in \{1,\ldots,K\}$.

**Proof:** $\quad \lim_{e\to\infty}(V_k\Lambda_k V_k^{-1})^e = \lim_{e\to\infty} V_k\Lambda_k^e V_k^{-1} = V_k \begin{bmatrix} 1 & 0 & \cdots & 0 \\ 0 & 0 & \cdots & 0 \\ & & \ddots & \\ 0 & 0 & \cdots & 0 \end{bmatrix} V_k^{-1} = v_k 1_M^\top.$

The third step follows because there is only one eigenvalue with magnitude 1. Since multiplying $\bar{A}$ by itself amounts to multiplying the corresponding diagonal blocks, we have the structure in (6). $\square$

Note that equation (6) points out that the matrix $\lim_{e\to\infty} \bar{A}^e$ consists of $K$ blocks of size $M \times M$ where the $k$'th block is $v_k 1_M^\top$. A straightforward algorithm can now be developed for making $\bar{A}^{\mathcal{P}}$ block diagonal. Since the eigenvalue decomposition is invariant under permutation, $\bar{A}$ and $\bar{A}^{\mathcal{P}}$ have the same eigenvalues and eigenvectors. As $e \to \infty$, $K$ clusters of columns appear in $(\bar{A}^{\mathcal{P}})^e$. Thus, $\bar{A}^{\mathcal{P}}$ can be made block-diagonal by clustering the columns of $(\bar{A}^{\mathcal{P}})^\infty$. This idea is illustrated in the middle row of Figure 1. Note that, in an actual implementation, one would use a low-rank reconstruction by zeroing-out the eigenvalues that are not equal to 1 in $\bar{\Lambda}$ to form $(\bar{A}^{\mathcal{P}})^r := \bar{V}^{\mathcal{P}}(\bar{\Lambda}^{\mathcal{P}})^r(\bar{V}^{\mathcal{P}})^{-1} = (\bar{A}^{\mathcal{P}})^\infty$, where $(\bar{\Lambda}^{\mathcal{P}})^r \in \mathbb{R}^{MK \times MK}$ is a diagonal matrix with only $K$ non-zero entries, corresponding to the eigenvalues which are 1.

This algorithm corresponds to the noiseless case $\bar{A}^{\mathcal{P}}$. In practice, the output of the learning algorithm is $\bar{A}_\epsilon^{\mathcal{P}}$ and the clear structure in Equation (6) no longer holds in $(\bar{A}_\epsilon^{\mathcal{P}})^e$, as $e \to \infty$, as illustrated in the bottom row of Figure 1. We can see that the three-cluster structure no longer holds for large $e$. Instead, the columns of the transition matrix converge to a global stationary distribution.

### 3.2.2 Estimating the permutation in the presence of noise

In the general case with noise $\epsilon$, we lose the spectral property that the global transition matrix has $K$ eigenvalues which are 1. Consequently, the algorithm described in Section 3.2.1 cannot be

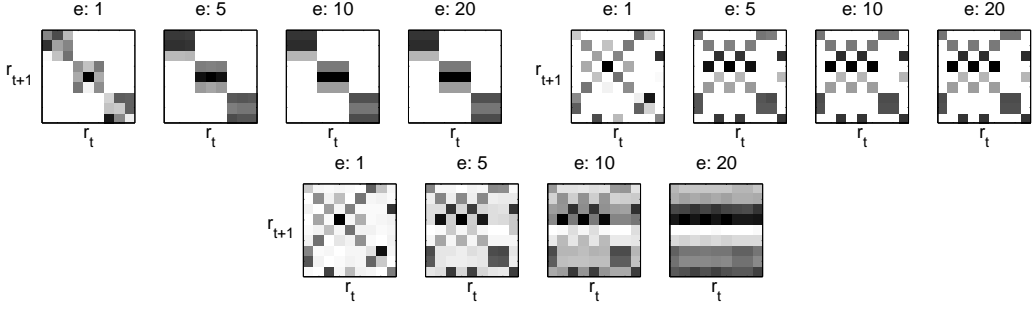

Figure 1: (Top left) Block-diagonal transition matrix after $e$-fold exponentiation. Each block converge to its own stationary distribution. (Top right) Same as above with permutation. (Bottom) Corrupted and permuted transition matrix after exponentiation. The true number $K = 3$ of HMMs is clear for intermediate values of $e$, but as $e \to \infty$, the columns of the matrix converge to a global stationary distribution.

applied directly to make $\bar{A}_\epsilon^{\mathcal{P}}$ block diagonal. In practice, the estimated transition matrix has only one eigenvalue with unit magnitude and $\lim_{e \to \infty}(\bar{A}_\epsilon^{\mathcal{P}})^e$ converges to a global stationary distribution. However, if the noise $\epsilon$ is sufficiently small, a depermutation mapping $\widetilde{\mathcal{P}}$ and the number of HMM clusters $K$ can be successfully estimated. We now specify the spectral conditions for this.

**Definition 1:** We denote $\lambda_k^{\mathcal{G}} := \alpha_k \lambda_{1,k}$ for $k \in \{1, \ldots, K\}$ as the global, noisy eigenvalues with $|\lambda_k^{\mathcal{G}}| \geq |\lambda_{k+1}^{\mathcal{G}}|, \forall k \in \{1, \ldots, K-1\}$, where $\lambda_{1,k}$ is the original eigenvalue of the $k^{\text{th}}$ cluster with magnitude 1 and $\alpha_k$ is the noise that acts on that eigenvalue (note that $\alpha_1 = 1$). We denote $\lambda_{j,k}^{\mathcal{L}} := \beta_{j,k} \lambda_{j,k}$ for $j \in \{2, \ldots, M\}$ and $k \in \{1, \ldots, K\}$ as the local, noisy eigenvalues with $|\lambda_{j,k}^{\mathcal{L}}| \geq |\lambda_{j+1,k}^{\mathcal{L}}|, \forall k \in \{1, \ldots, K\}$ and $\forall j \in \{1, \ldots, M-1\}$, where $\lambda_{j,k}$ is the original eigenvalue with the $j^{\text{th}}$ largest magnitude in the $k^{\text{th}}$ cluster, and $\beta_{j,k}$ is the noise that acts on that eigenvalue.

**Definition 2:** The low-rank eigendecomposition of the estimated transition matrix $\bar{A}_\epsilon^{\mathcal{P}}$ is defined as $A_\epsilon^r := V \Lambda^r V^{-1}$, where $V$ is a matrix with eigenvectors in the columns and $\Lambda^r$ is a diagonal matrix with eigenvalues $\lambda_{1:K}^{\mathcal{G}}$ in the first $K$ entries.

**Conjecture 1:**

If $|\lambda_K^{\mathcal{G}}| > \max_{k \in \{1,\ldots,K\}} |\lambda_{2,k}^{\mathcal{L}}|$, then $A^r$ can be formed using the eigen-decomposition of $\bar{A}_\epsilon^{\mathcal{P}}$. Then, with high probability, $\|A_\epsilon^r - A^r\|_F \leq \mathcal{O}(1/\sqrt{TN})$, where $TN$ is the total number of observed vectors.

**Justification:**

$$
\begin{aligned}
\|A_\epsilon^r - A^r\|_F = \|A_\epsilon^r - A + A - A^r\|_F &\leq \|A_\epsilon^r - A\|_F + \|A - A^r\|_F \\
&= \|A - A^r\|_F + \|A - A_\epsilon + A_\epsilon^{\bar{r}}\|_F \\
&\leq \|A - A^r\|_F + \|A_\epsilon^{\bar{r}}\|_F + \|A - A_\epsilon\|_F \\
&\leq 2KM + \mathcal{O}(1/\sqrt{TN}) = \mathcal{O}(1/\sqrt{TN}), \quad w.h.p.,
\end{aligned}
$$

where $A$ is used for $\bar{A}^{\mathcal{P}}$ to reduce the notation clutter (and similarly $A^r$ for $(\bar{A}^{\mathcal{P}})^r$ and so on), we used the triangle inequality for the first and second inequalities and $A_\epsilon^{\bar{r}} = V \Lambda^{\bar{r}} V^{-1}$, where $\Lambda^{\bar{r}}$ is a diagonal matrix of eigenvalues with the first $K$ diagonal entries equal to zero (complement of $\Lambda^r$). For the last inequality, we used the fact that $A \in \mathbb{R}^{MK \times MK}$ has entries in the interval $[0, 1]$ and we used the sample complexity result from [1]. The bound specified in [1] is for a mixture model, but since the two models are similar and the estimation procedure is almost identical, we are reusing it. We believe that further analysis of the spectral learning algorithm is out of the scope of this paper, so we leave this proposition as a conjecture. $\qquad \square$

Conjecture 1 asserts that, if we have enough data we should obtain an estimate $A_\epsilon^r$ close to $A^r$ in the squared error sense. Furthermore, if the following mixing rate condition is satisfied, we will be able to identify the number of clusters $K$ from the data.

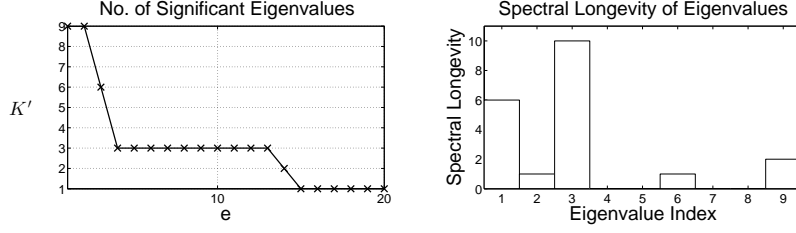

Figure 2: (Left) Number of significant eigenvalues across exponentiations. (Right) Spectral Longevity $\mathfrak{L}_{\tilde{\lambda}_{K'}}$ with respect to the eigenvalue index $K'$.

**Definition 3:** Let $\widetilde{\lambda}_k$ denote the $k^{\text{th}}$ largest eigenvalue (in decreasing order) of the estimated transition matrix $\bar{A}_\epsilon^{\mathcal{P}}$. We define the quantity,

$$
\mathfrak{L}_{\tilde{\lambda}_{K'}} := \sum_{e=1}^{\infty} \left( \left[ \frac{\sum_{l=1}^{K'} |\tilde{\lambda}_l|^e}{\sum_{l'=1}^{MK} |\tilde{\lambda}_{l'}|^e} > 1-\gamma \right] - \left[ \frac{\sum_{l=1}^{K'-1} |\tilde{\lambda}_l|^e}{\sum_{l'=1}^{MK} |\tilde{\lambda}_{l'}|^e} > 1-\gamma \right] \right), \tag{7}
$$

as the *spectral longevity* of $\tilde{\lambda}_{K'}$. The square brackets [.] denote an indicator function which outputs 1 if the argument is true and 0 otherwise, and $\gamma$ is a small number such as machine epsilon.

**Lemma 3:** If $|\lambda_K^{\mathcal{G}}| > \max_{k \in \{1,\ldots,K\}} |\lambda_{2,k}^{\mathcal{L}}|$ and $\arg\max_{K'} \frac{|\tilde{\lambda}_{K'}|^2}{|\tilde{\lambda}_{K'+1}||\tilde{\lambda}_{K'-1}|} = K$, for $K' \in \{2, 3, \ldots, MK-1\}$, then $\arg\max_{K'} \mathfrak{L}_{\tilde{\lambda}_{K'}} = K$.

**Proof:** The first condition ensures that the top $K$ eigenvalues are global eigenvalues. The second condition is about the convergence rates of the two ratios in equation (7). The first indicator function has the following summation inside:

$$
\frac{\sum_{l=1}^{K'} |\tilde{\lambda}_l|^e}{\sum_{l'=1}^{MK} |\tilde{\lambda}_{l'}|^e} = \frac{\sum_{l=1}^{K'-1} |\tilde{\lambda}_l|^e + |\tilde{\lambda}_{K'}|^e}{\sum_{l'=1}^{K'-1} |\tilde{\lambda}_{l'}|^e + |\tilde{\lambda}_{K'}|^e + |\tilde{\lambda}_{K'+1}|^e + \sum_{l'=K'+2}^{MK} |\tilde{\lambda}_{l'}|^e} \ .
$$

The rate at which this term goes to 1 is determined by the spectral gap $|\lambda_{K'}|/|\lambda_{K'+1}|$. The smaller this ratio is, the faster the term (it is non-decreasing w.r.t. $e$) converges to 1. For the second indicator function inside $\mathfrak{L}_{\tilde{\lambda}_{K'}}$, we can do the same analysis and see that the convergence rate is again determined by the gap $|\tilde{\lambda}_{K'-1}|/|\tilde{\lambda}_{K'}|$. The ratio of the two spectral gaps determines the spectral longevity. Hence, for the $K'$ with largest ratio $\frac{|\tilde{\lambda}_{K'}|^2}{|\tilde{\lambda}_{K'+1}||\tilde{\lambda}_{K'-1}|}$, we have $\arg\max_{K'} \mathfrak{L}_{\tilde{\lambda}_{K'}} = K$. $\square$

Lemma 3 tells us the following. If the estimated transition matrix $\bar{A}_\epsilon^{\mathcal{P}}$ is not *too noisy*, we can determine the number of clusters by choosing the value of $K'$ such that it maximizes $\mathfrak{L}_{\tilde{\lambda}_{K'}}$. This corresponds to exponentiating the sorted eigenvalues in a finite range, and recording the number of non-negligible eigenvalues. This is depicted in Figure 2.

### 3.3 Proposed Algorithm

In previous sections, we have shown that the permutation caused by the MoM estimation procedure can be undone, and we have proposed a way to estimate the number of clusters $K$. We summarize the whole procedure in Algorithm 1.

## 4 Experiments

### 4.1 Effect of noise on depermutation algorithm

We have tested the algorithm's performance with respect to amount of data. We used the parameters $K = 3$, $M = 4$, $L = 20$, and we have 2 sequences with length $T$ for each cluster. We used a Gaussian observation model with unit observation variance and the columns of the emission matrices $O_{1:K}$ were drawn from zero mean spherical Gaussian with variance 2. Results for 10 uniformly

---

**Algorithm 1** Spectral Learning for Mixture of Hidden Markov Models

---

**Inputs:** $\mathbf{x}_{1:N}$ : Sequences, $MK$ : total number of states of global HMM.

**Output:** $\widehat{\theta} = \left( \widehat{O}_{1:\widehat{K}}, \widehat{A}_{1:\widehat{K}} \right)$ : MHMM parameters

**Method of Moments Parameter Estimation**

   $(\bar{O}_\epsilon^{\mathcal{P}}, \bar{A}_\epsilon^{\mathcal{P}}) = \text{HMM\_MethodofMoments}(\mathbf{x}_{1:N}, MK)$

**Depermutation**

   Find eigenvalues of $\bar{A}_\epsilon^{\mathcal{P}}$

   Exponentiate eigenvalues for each discrete value $e$ in a sufficiently large range.

   Identify $\widehat{K}$ as the eigenvalue with largest longevity.

   Compute rank-$\widehat{K}$ reconstruction $A_\epsilon^r$ via eigendecomposition.

   Cluster the columns of $A_\epsilon^r$ with $\widehat{K}$ clusters to find a depermutation mapping $\widetilde{\mathcal{P}}$ via cluster labels.

   Depermute $\bar{O}_\epsilon^{\mathcal{P}}$ and $\bar{A}_\epsilon^{\mathcal{P}}$ according to $\widetilde{\mathcal{P}}$.

   Form $\widehat{\theta}$ by choosing corresponding blocks from depermuted $\bar{O}_\epsilon^{\mathcal{P}}$ and $\bar{A}_\epsilon^{\mathcal{P}}$.

**Return** $\widehat{\theta}$.

---

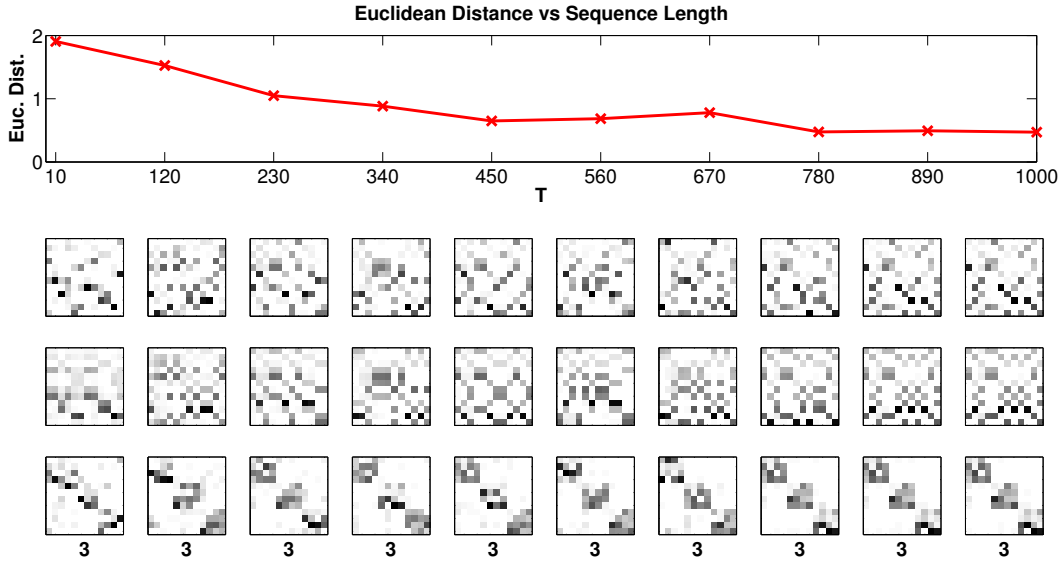

Figure 3: Top row: Euclidean distance vs $T$. Second row: Noisy input matrix. Third row: Noisy reconstruction $A_\epsilon^r$. Bottom row: Depermuted matrix, numbers at the bottom indicate the estimated number of clusters.

spaced sequence lengths from 10 to 1000 are shown in Figure 3. On the top row, we plot the total error (from centroid to point) obtained after fitting k-means with true number of HMM clusters. We can see that the correct number of clusters $K = 3$ as well as the block-diagonal structure of the transition matrix is correctly recovered even in the case where $T = 20$.

## 4.2 Amount of data vs accuracy and speed

We have compared clustering accuracies of EM and our approach on data sampled from a Gaussian emission MHMM. Means of each state of each cluster is drawn from a zero mean unit variance Gaussian, and observation covariance is spherical with variance 2. We set $L = 20$, $K = 5$, $M = 3$. We used uniform mixing proportions and uniform initial state distribution. We evaluated the clustering accuracies for 10 uniformly spaced sequence lengths (every sequence has the same length) between 20 and 200, and 10 uniformly spaced number of sequences between 1 and 100 for each cluster. The results are shown in Figure 4. Although EM seems to provide higher accuracy on

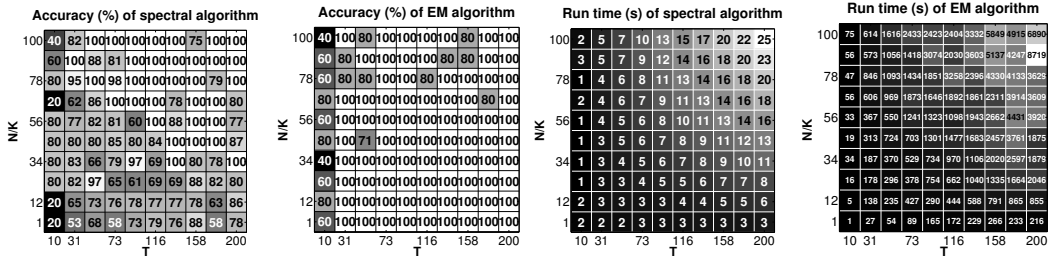

Figure 4: Clustering accuracy and run time results for synthetic data experiments.

Table 1: Clustering accuracies for handwritten digit dataset.

| Algorithm | 1v2 | 1v3 | 1v4 | 2v3 | 2v4 | 2v5 |
|---|---|---|---|---|---|---|
| Spectral | 100 | 70 | 54 | 83 | 99 | 99 |
| EM init. w/ Spectral | 100 | 99 | 100 | 96 | 100 | 100 |
| EM init. at Random | 96 | 99 | 98 | 83 | 100 | 100 |

regions where we have less data, spectral algorithm is much faster. Note that, in spectral algorithm we include the time spent in moment computation. We used four restarts for EM, and take the result with highest likelihood, and used an automatic stopping criterion.

## 4.3 Real data experiment

We ran an experiment on the handwritten character trajectory dataset from the UCI machine learning repository [8]. We formed pairs of characters and compared the clustering results for three algorithms: the proposed spectral learning approach, EM initialized at random, and EM initialized with MoM algorithm as explored in [9]. We take the maximum accuracy of EM over 5 random initializations in the third row. We set the algorithm parameters to $K = 2$ and $M = 4$. There are 140 sequences of average length 100 per class. In the original data, $L = 3$, but to apply MoM learning, we require that $MK < L$. To achieve this, we transformed the data vectors with a cubic polynomial feature transformation such that $L = 10$ (this is the same transformation that corresponds to a polynomial kernel). The results from these trials are shown in Table 1. We can see that although spectral learning doesn't always surpass randomly initialized EM on its own, it does serve as a very good initialization scheme.

## 5   Conclusions and future work

We have developed a method of moments based algorithm for learning mixture of HMMs. Our experimental results show that our approach is computationally much cheaper than EM, while being comparable in accuracy. Our real data experiment also show that our approach can be used as a good initialization scheme for EM. As future work, it would be interesting to apply the proposed approach on other hierarchical latent variable models.

**Acknowledgements:** We would like to thank Taylan Cemgil, David Forsyth and John Hershey for valuable discussions. This material is based upon work supported by the National Science Foundation under Grant No. 1319708.

## Footnotes

[1]Without loss of generality, the number of hidden states for each HMM is taken to be $M$ to keep the notation uncluttered.

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
