[Reviews · NeurIPS 2014]

Submitted by Assigned_Reviewer_4

Summary: The authors consider the problem of learning a mixture of Hidden Markov Models. The authors first suggest using a spectral learning algorithm to learn a set of parameters for a hidden Markov model, and then provide a method for resolving the permutation ambiguity in the transition matrix to recover it’s underlying block-diagonal structure.

Major Comments:

1. I found this paper to be very well written for the most part. The experimental results section could be fleshed out a bit. In particular the

2. The authors rely on the fact that a mixture of Hidden Markov Models can be expressed as a single HMM. One of the main reasons, if not the primary reason, for learning a mixture of HMMs as opposed to a single HMM directly is that the block diagonal structure of the mixture of HMMs is sparse. This sparseness may be imposed on problems that are deemed to be too large to learn with a full model, especially if the problem is large and the amount of training data is small. However, in this paper, the author’s algorithm relies on *first* accurately estimating the full HMM parameters, then reversing the permutation in order to find the block diagonal structure. For most real-world problems this seems completely untenable. Do the author’s have suggestions for when their algorithm might be appropriate and when it might not be?

3. Again, in practice, the recovered transition matrix is not likely to be sparse. Is there any way to guarantee that the transition matrix learned by the spectral algorithms suggested in this paper will be (close to) sparse? I suspect that there may be many HMM models with full transition matrices that are close to the sparse model with block diagonal structure that the authors are looking for. It is not at all obvious that the matrix that is learned via [1,2] will be sparse with very small amounts of noise.

4. I am really worried that the experimental results do not accurately reflect the reality of noise in the transition matrix. For example, in Section 4.1, the authors apply their algorithm to a transition matrix that has been permuted by noise sampled from a Dirichlet distribution… but this is a paper about learning mixtures of HMM parameters, not just depermuting matrices. Shouldn’t the authors have sampled observation data from a mixture of HMMs and then learned all of the parameters back from the *observed* data? This would give a much more accurate idea of how noise in observations and small quantities of data are actually reflected in the learned transition matrix, and how hard it is to depermute the matrix in the presence of this noise.

5. The experiment in Section 4.2 is starting to get at what I was hoping for in the previous section, but there is nowhere near enough information to evaluate the quality of the experiment. In particular, I have no idea what the mixture of HMMs actually looked like. The actual parameters of the transition and observation matrix matter *a lot* for evaluating how easy or hard it is to learn the model from data. I would like to know more about the specifics of these parameters (the authors could put it in an appendix if they feel that space is at a premium).

6. In table 1, why is EM initialized with spectral learning ever doing worse than spectral learning alone? Shouldn’t EM only improve the results?

Minor Comments:

1. I believe that the subscript of \Lambda in the bottom right corner of the first matrix in the proof in Section 3.2.1 should be K, not 2.

2. Text and numbers are *way* too small in the figures.
Summary: The paper has some interesting ideas and good theoretical results. I am worried that the method is less likely to work on real datasets in practice.

Submitted by Assigned_Reviewer_20

This paper presents a spectral algorithm for learning a mixture of hidden Markov models (MHMM). It shows that an existing spectral algorithm for learning HMMs (Anandkumar et al., 2012) can be used off-the-shelf by formulating an MHMM as a special case of HMM. It proposes a stabilization-based solution for parameter permutation by introducing an assumption that each HMM's transition matrix has a single eigenvalue of value 1.

There seem to be some issues with notation. The authors definitely need to show more awareness of related work by providing references (e.g., initializing EM with spectral estimates is already explored in Chaganty and Liang (2013)). Experiments are quite basic, and I strongly recommend reporting results in a table format rather than the graphical format in Figure 4.

That said, the proposed scheme to dodge the problem of parameter permutation is interesting and intuitive. The authors also provide an analysis for the case with estimation noise, which leads to an intriguing usage of eigenvalues (Equation (7)) in the algorithm.

Comments:

- I was initially confused by the terms in Corollary 1. It'd be good to define 1_J^T (as a row vector of length J filled with ones) and elaborate a bit on the structure of lim_{e->inf} bar{A}^e (that it's JK x JK block diagonal where the columns within each J x J block are the same).

- Generally, please define each vector/matrix along with its dimensions when it's first introduced. It will immensely help the presentation.

- I find Figure 4 very hard to read: please replace with tables. Also, I'm not sure why EM is performing so badly in Table 1. Typically, even with random initialization, with enough iterations it beats spectral methods (despite the local optima problem). A standard practice is to report the best (not average) accuracy of EM: please include that result.

- Nit: It might help readers if the paper conforms to the established notation of Hsu et al. (2009). Namely, use T for transition matrices (not A), use n, m for numbers of observation and hidden states (not L, J).
Summary: The paper proposes an algorithm for learning a mixture of hidden Markov models. This algorithm consists of an existing spectral algorithm for learning hidden Markov models followed by a novel stabilization step for undoing parameter permutation.

Submitted by Assigned_Reviewer_36

The authors present spectral method for learning a mixture of HMMs.
The algorithm would be a trivial variant of the standard Hsu, Kakade & Zhang
algorithm, except that the transition matrix recovered is an arbitrary permutation of
the true one, mixing the various components. The authors propose a depermutation algorithm
that seems to work reasonably well.

Occasionally there seem to be a number or little typos. e.g.:
"For each e, identify of eigenvalues that contain all of the energy."
Please proof read more carefully.

The paper could do a better job of putting this work into context, perhaps relating to papers such as:
Chaganty and Liang. Spectral experts for estimating mixtures of linear regressions. In
International Conference on Machine Learning (ICML), 2013.
Summary: The authors present spectral method for learning a mixture of HMMs, addressing the key question of
who to "de-permute" the results.
Author Feedback
Author rebuttal: We thank all three reviewers for their comments.

Regarding all visual, notational and explanatory concerns, we agree with the reviewers. We will make the text in the figures larger, fix typos, and add the suggested explanations. We agree that using the the established notation in Hsu et al. 2009 paper will be helpful to the reader. Chaganty and Liang’s 2013 work will definitely be added to the final manuscript. We will convert Figure 4 to a table format to make it easier to read.

Regarding the concerns about the EM performance in table 1, we see that if we take the highest accuracy among random initializations (instead of the average as we previously did), randomly initialized EM equals/beats the performance of spectral learning alone as Reviewer 20 suggests. This will be included in the final version of the paper. Furthermore, as pointed out by Reviewer 4, EM was decreasing the accuracy obtained with spectral learning. We have found the reason that was causing this problem: We were feeding random initial cluster indicators to EM alongside the parameters learned with spectral learning. After rectifying this, EM initialized with spectral learning performs better than or the same as spectral learning alone, as expected. The revised results will be included in the paper to alleviate such confusions.

Reviewer 4 asks whether the overall algorithm would perform well under real data circumstances, which may ruin the sparsity of the global transition matrix estimated with spectral learning algorithms in [1,2]:
For problems where the data is modelled accurately enough with an MHMM (model mismatch is not too serious), dimensionality of the observations (L) is large enough, and there is enough data, the spectral learning algorithms in [1,2] returns sparse enough solutions to be able to recover the block-diagonal structure of the global transition matrix, and therefore the proposed algorithm is appropriate to use. As also can be seen from Figure 3, even when the input matrix is not very sparse when alpha=0.4, the depermutation algorithm is successful. That being said, we agree that imposing sparsity in the estimation step of the algorithm would be a great help to increase the robustness of the overall approach. In this paper, we have not investigated how to impose sparsity in estimation step, but we do think that it is a very interesting future research direction.

Reviewer 4 states that in Figure 3 it would be better to use transition matrices estimated with [1,2] to better understand how does the algorithm perform under actual estimation noise:
We have observed that with synthetic data where there is no model mismatch, the estimation noise tends to be sparse and is most of the time tolerable by the proposed depermutation algorithm. We used uniform Dirichlet noise in Figure 3 to demonstrate that the proposed depermutation algorithm is able to work under non-sparse noise. We will add the suggested figure to the final version.

Finally, in response to reviewer 4’s inquiry about the experiment details in Section 4.2:
We use a 20 dimensional (L=20) Gaussian emission model with fixed variance (fixed to be one) and mean of each state of each cluster is drawn from a zero mean Gaussian with unit variance. The columns of the transition matrix of each cluster are drawn from a Dirichlet (1,...,1) distribution. We have 3 clusters (K=3) and each HMM has 3 states (J=3). We have uniform mixing proportions, and uniform initial states. We will include these details in the final manuscript.